# Mixup Augmentation for Kidney and Kidney Tumor Segmentation

Matej Gazda[1], Peter Bugata[1], Jakub Gazda[2], David Hubacek[3], David Jozef Hresko[1], and Peter Drotar[1]

[1] IISlab, Technical University of Kosice, Slovakia
[2] 2nd Department of Internal Medicine, Pavol Jozef Safarik University and Louis Pasteur University Hospital, Kosice, Slovakia
[3] Department of Radiodiagnostics and Medical Imaging, Louis Pasteur University Hospital, Kosice, Slovakia
matej.gazda@tuke.sk

**Abstract.** Abdominal computed tomography is frequently used to non-invasively map local conditions and to detect any benign or malign masses. However, ill-defined borders of malign objects, fuzzy texture, and time pressure in fact, make accurate segmentation in clinical settings a challenging task. In this paper, we propose a two-stage deep learning architecture for kidney and kidney masses segmentation, denoted as convolutional computer tomography network (CCTNet). The first stage locates volume bounding box containing both kidneys. The second stage performs the segmentation of kidney, kidney tumors and cysts. In the first stage, we use a pre-trained 3D low resolution nnU-Net. In the second stage, we employ a mixup augmentation to improve segmentation performance of the second 3D full resolution nnU-Net. The obtained results indicate that CCTNet can provide improved segmentation of kidney, kidney tumor and cyst.

**Keywords:** nnU-Net · mixup · kidney · CT · segmentation .

## 1 Introduction

The kidney cancer is one of the ten most common cancers, and it is the third most common genitourinary malignancy [3] with high mortality. Kidney cysts, although being benign, are also a cause for concern due to their potential for malign transformation. Thus, it is essential to maximize the rate of their diagnosis, especially in early stages, when curative treatment is still possible. Medical imaging, such as MRI, CT and US, plays crucial role in detecting both kidney cyst and kidney cancer. Nowadays, kidney cancer cases are frequently found only incidentally during the regular medical image evaluation [4].

The incidence of renal cell carcinoma (RCC; one of the most common kidney cancers) is continuously growing and, since the 1990-ies, the incidence has doubled in developed world [4]. This may be attributed to advances in medical imaging technology, which provide more precise and detailed images than ever

before, and to a more frequent employment of imaging techniques in everyday clinical practice. RCC currently accounts for more than 400,000 new cases worldwide every year [4]. It affects mainly the population of those older than 60 years and thus, with the aging world population, the number of patients is expected to increase even further.

Early diagnosis and subsequent treatment of kidney cancer is critical because it improves both health-related quality of life and overall survival. Development of computer-assisted diagnosis is important since it can help to automate MRI/CT/US evaluation, release some workload burden from radiologists, increase the number of evaluated images and in turn increase the proportion of RCC patients diagnosed in the very early stages, when curative treatment may be possible.

Currently, state-of-the-art models for semantic segmentation are based on U-net architecture, V-net architecture or their 3D derivatives. These architectures were also backbone for the recent Kidney and Kidney Tumor Segmentation Challenge 2019 (KITS 2019) [1]. The best performing models in final results were based on U-net and V-net architectures. The ultimate winner of KITS 2019 challenge nnU-net [2] proved itself also on several other segmentation challenges and currently represents gold standard in medical image semantic segmentation.

In this paper, we build upon the nnU-net and propose to use mixup [6] augumentation that was shown to improve the generalization of the state-of-the-art neural network architectures. The results obtained on published KITS2019 dataset indicate that mixup can be with advantage used also in segmentation of 3D CT images. We achieved improved performance on all evaluated classes.

In the next section, we provide the description of the proposed architecture together with the details of network training and validation. Then, the results of the proposed network and the baseline are given, followed by discussion.

## 2   Methods

Previous results indicate [2] that most of the performance gains are not network architecture dependent, but rather can be obtained by careful tuning of the network parameters. Another part of the machine learning pipeline that yields performance improvements is the data pre-processing and post-processing. As a such we did not attempt to find the architecture modifications to boost the performance but rather try to provide better condition for network training through mixup augumentation [6].

### 2.1   Training and Validation Data

Our submission made use of the official KiTS21 training set alone and we used majority voting aggregation for multiple segmentation annotations.

## 2.2   Preprocessing

The data transformation, re-sampling, and normalisation were handled by the nnU-Net configuration. As described in [2], this means for anisotropic data that image resampling strategy in place was third order spline interpolation for in-plane and nearest neighbours for out-of-plane. The normalisation was global dataset percentile clipping and z-score with global foreground mean and s.d. The clipping of HU values was handled by nnU-net default setting (0.5 and 99.5 percentile).

Since wrong labels can limit the quality of predictions learned by deep neural networks [5], we analysed true labels provided by the challenge organizers. Three cases were modified after the consultation with radiologist. In the case of 084, the mass structure attached to the left kidney is labeled as kidney. We cleared the kidney label, however we did not add any label for the mass, since it was difficult to decide whether this is a cyst or a tumor. The another modified case, 277, was more tricky. The labels provided by the organisers denote the cyst and tumor in the left kidney. However, closer inspection shows that this is in fact a single structure. In this case, we replaced the cyst label by the tumor label. Finally, in the third modified case - 299, we corrected the label for the left kidney. Axial view showed some nodule growing out of the left kidney, that is labeled as kidney, but has different density and does not fit to the kidney shape. We removed the kidney label for this nodule.

## 2.3   Proposed Method

The proposed classifier network consists of two cascade connected nnU-Net networks as depicted in Fig. 1. In the first stage we employ the nnU-Net to locate kidney in 3D CT image and crop volume containing kidneys. This step has two goals. To reduce computational cost in the second stage, and to eliminate possible erroneous predictions (e.g. cyst lying in some distant areas of CT image). Since the segmentation achieved by nnU-Net in KITS 2019 challenge was sufficiently high, we take advantage of the nnU-Net pre-trained on data from KITS 2019 challenge. To avoid some errors on the borders, we add 60 voxels in every direction to the cropped volume containing the kidneys.

To train nnU-Net in the second stage we use mixup [6] augmentation. Mixup is a data-agnostic augmentation routine that constructs virtual training examples by convex combinations of pairs of training examples and their labels according to the rule [6]

$$x_{mixup} = \lambda x_i + (1 - \lambda)x_j, \tag{1}$$

$$y_{mixup} = \lambda y_i + (1 - \lambda)y_j, \tag{2}$$

where $(x_i, y_i)$ and $(x_j, y_j)$ are two random training data samples, and $\lambda \in [0, 1]$. The $\lambda$ is distributed according Beta distribution $\lambda \sim \beta(\alpha, \alpha)$, where $\alpha \in (0, \infty)$.

The networks parameters are automatically configured by nnU-Net, meaning that as a loss function a combination of Dice and Cross-entropy loss is used, and SGD with Nesterov momentum is used for network training optimization. To validate our approach, we used 5-fold cross-validation.

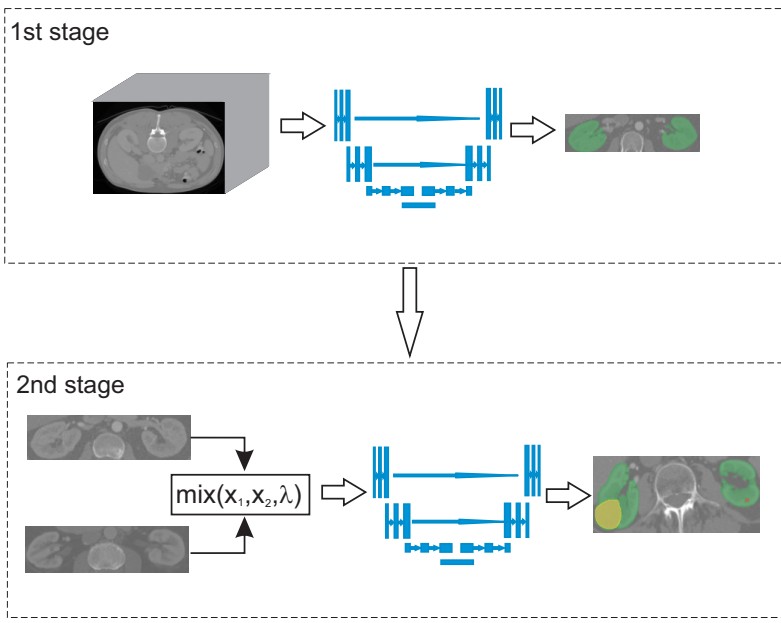

Fig. 1: Proposed two stage approach (CCTNet).

## 3    Results

In order to objectively asses the performance of the proposed approaches, three Hierarchical Evaluation Classes (HECs) were introduced : kidney and masses, kidney mass, and tumor. Kidney and masses HEC includes semantic classes kidney, tumor, and cyst. Kidney mass HEC covers only tumor and cyst. Finally, tumor HEC is the same as the semantic class for tumor.

To train the final model, we used batch size equal to four and the mixup augmentation with $\lambda \sim \beta(\alpha = 0.4)$.

The results in terms of combined Sorensen-Dice loss and Surface Dice loss are presented in Tab. 2. We provide results for HECs as well as individual semantic classes. As can be seen, the CCTNet provides improvement for each of considered hierarchical classes. This is also confirmed by evaluating the performance on individual semantic classes. By closer analysis of the results, we discovered that this improvement comes not only from mixup augmentation alone. We also noticed an improvement after the implementation of the crop in the first stage.

The Fig. 2 shows the example prediction of the CCTNet. We randomly selected case 0193 from cases containing all three labels (kidney, tumor, cyst). As can be seen, the segmentation of kidneys is pretty good. There is some difference between prediction and ground truth in the cyst located in the right kidney. The most noticeable difference is visible in the tumor label. However, when comparing predictions to the unlabeled CT, we can notice that indeed the real tumor lies outside the ground truth annotated region.

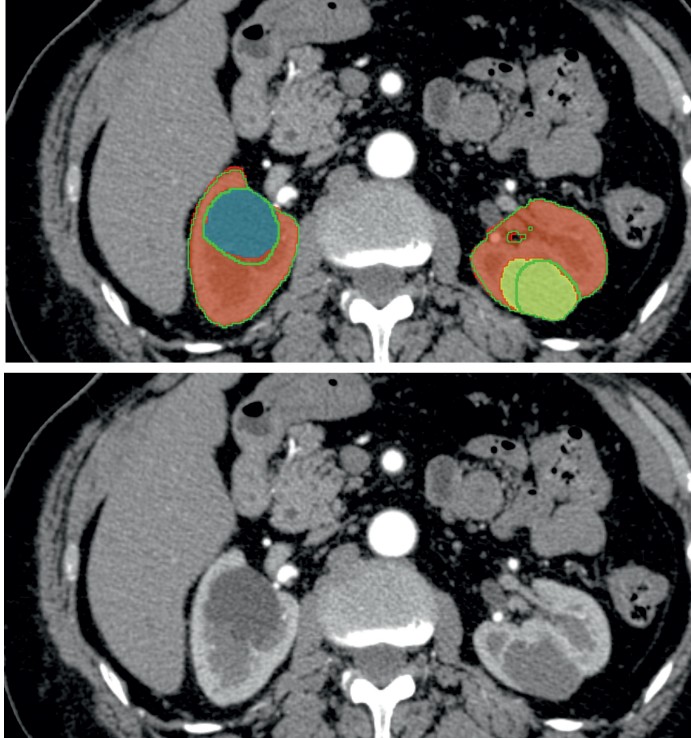

Fig. 2: Example segmentation for the 0193 case. Green contours denote ground truth label. Predicted segmentation classes are: red=kidney, yellow=tumor, blue=cyst.

Table 1: Detailed specification of 3D convolutional layers (conv) and transposed convolutions (T conv). Every conv layer is followed by normalisation and leaky ReLU activation function.

| Network | layer no | kernel | stride | Output dimension | Network | layer no | kernel | stride | Output dimension |
|---|---|---|---|---|---|---|---|---|---|
| conv | 0 | (1,3,3) | (1,1,1) | (56,256,128) | conv | 5 | (3,3,3) | (1,1,1) | (7,8,4) |
| conv | 0 | (1,3,3) | (1,1,1) | (56,256,128) | conv | 5 | (3,3,3) | (1,1,1) | (7,8,4) |
| conv | 1 | (3,3,3) | (1,2,2) | (56,256,64) | T conv | 5 | (1,2,2) | (1,2,2) | (7,16,8) |
| conv | 1 | (3,3,3) | (1,1,1) | (56,256,64) | conv | 4 | (3,3,3) | (1,1,1) | (7,16,8) |
| conv | 2 | (3,3,3) | (2,2,2) | (28,64,32) | conv | 4 | (3,3,3) | (1,1,1) | (7,16,8 |
| conv | 2 | (3,3,3) | (1,1,1) | (28,64,32) | T conv | 4 | (2,2,2) | (2,2,2) | (14,32,16) |
| conv | 3 | (3,3,3) | (2,2,2) | (14,32,16) | conv | 3 | (3,3,3) | (1,1,1) | (14,32,16) |
| conv | 3 | (3,3,3) | (1,1,1) | (14,32,16) | conv | 3 | (3,3,3) | (1,1,1) | (14,32,16) |
| conv | 4 | (3,3,3) | (2,2,2) | (7,16,8) | T conv | 3 | (2,2,2) | (2,2,2) | (28,64,32) |
| conv | 4 | (3,3,3) | (1,1,1) | (7,16,8) | conv | 2 | (3,3,3) | (1,1,1) | (28,64,32) |
| conv | 5 | (3,3,3) | (1,2,2) | (7,8,4) | conv | 2 | (3,3,3) | (1,1,1) | (28,64,32) |
| conv | 5 | (3,3,3) | (1,1,1) | (7,8,4) | T conv | 2 | (2,2,2) | (2,2,2) | (56,128,64) |
| | | | | | conv | 1 | (3,3,3) | (1,1,1) | (56,128,64) |
| | | | | | conv | 1 | (3,3,3) | (1,1,1) | (56,128,64) |
| | | | | | T conv | 1 | (1,2,2) | (1,2,2) | (56,256,128) |
| | | | | | conv | 0 | (3,3,3) | (1,1,1) | (56,256,128) |
| | | | | | conv | 0 | (3,3,3) | (1,1,1) | (56,256,128) |
| | | bottleneck - conv | 6 | (3,3,3) | (1,2,1) | (7,4,4) | | | |
| | | bottleneck - conv | 6 | (3,3,3) | (1,1,1) | (7,4,4) | | | |

Table 2: Average of Sorensen-Dice loss and Surface Dice loss for predicted hierarchical and semantic classes

| | HECs | | | semantic classes | | |
|---|---|---|---|---|---|---|
| network | Kidney and Masses | Kidney Mass | Tumor | Kidney | Tumor | Cyst |
| CCTNet (ours) | 0.00896 | 0.00518 | 0.00236 | 0.04580 | 0.01100 | 0.01026 |
| 3D full res nnU-net (baseline) | 0.01068 | 0.00730 | 0.00302 | 0.08090 | 0.02238 | 0.01804 |

## 4   Discussion

We have investigated errors in segmentation of kidney and masses to get a better insight on the network performance. In case of kidney, many errors were at the kidney's boundary. It was interesting to see that in some cases the network provided more accurate boundary than human annotator. However, in evaluation this is considered as a mistake since these two labels do not match.

As was already mentioned, the network tend to mark small cysts where they do not exist. There were also opposite cases, i.e., the network missed to find small cysts. This can be probably expected since these cysts are rather small and the volume is quite heterogeneous, so small cysts are hard to spot.

We will take a closer look at two erroneous predictions depicted in Fig. 3. As can be seen in Fig. 3a, the network marked part of the cyst (brown) as a tumor (yellow). So even though there exists only one single segment (cyst), the network concatenated two different segments (cyst and tumor). This also happens in several other predicted cases. During fault cases analyses, we identified a repeated pattern in true labels, where two distinct segments (cyst and tumor) share common boundaries and therefore create an illusion of one single object (approximately 10-15% of all cysts shared common boundaries with tumor). Therefore, we hypothesize that the network learned this feature and transformed it erroneously to predictions, even though cysts were completely benign with no malign characteristics.

The case depicted in Fig. 3b shows hydronephrosis segmented as a cyst. Hydronephrosis means renal pelvis enlargement, which can resemble kidney cyst on CT scans, when only nephrogenic phase is considered (period when all of the healthy renal parenchyma is contrast-enhanced). In practice, however, hydronephrosis and cysts are two distinct objects and radiologists need to consider also delayed phase to definitely distinguish between those two. However, this was rather a solitary case hardly with any real impact on network learning.

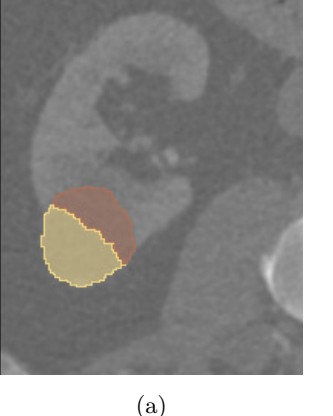
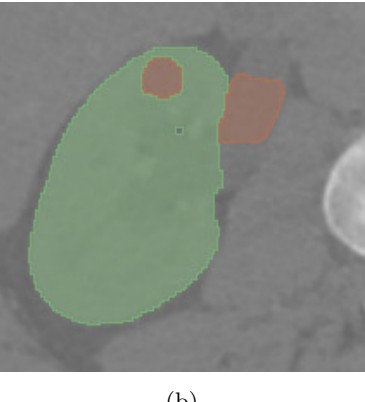

(a)                                    (b)

Fig. 3: Examples of incorrect segmentation of the proposed network: red represents cysts and yellow tumors.

We mentioned just few aspects, but more detailed investigation can reveal more similar cases. Some of these may be corrected by post-processing but for others some updates in network learning would be necessary.

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
