# OpenReview forum: "Mixup Augmentation for Kidney and Kidney Tumor Segmentation"
_MICCAI.org/2021/Challenge/KiTS — Submitted to KiTS21 Challenge_

### Official Review · Reviewer_r2kj · 2021-08-30

**Rating:** 6

**Review:**

The authors present a coarse-to-fine approach that makes use of mixup augmentation on top of the nnU-Net baseline that was provided. In general the paper is well-written and does a good job explaining what was done, but only a very brief results section is included with nothing but a single table. This should be significantly expanded to include some narrative and possible a figure. Also, the authors do not mention how they aggregated the multiple annotations per case into segmentations to use for training and validation. Most teams used majority voting -- if this is the case, they should state it explicitly.

---

### Official Review · Reviewer_N3Uf · 2021-08-30

**Rating:** 7

**Review:**

### Overall

- An institutional email address is preferred if possible

### Introduction

- I believe "automatize" should be "automate"
- I believe "derivations" should be "derivatives"

### Methods

- What percentiles were HU values clipped at?
- It would be great if you could share within your paper which cases you decided to modify and why

### Results

- Please expand on this method. Did you try the baseline alone? If so, what were your results and how did they compare to your method?
- It would be nice if you could include a figure that shows one of your predictions compared to the ground truth
- Once the final results are known, please add them to this section
- Please clarify whether these numbers are Dice or Dice Loss -- i.e. is higher better or is lower better? Since the official challenge will evaluate based on Dice (higher -> better) it would be preferable to also use that

### Discussion and Conclusion

- Nice work on this. I like how some error modes are discussed

---

### Decision · Program_Chairs · 2021-08-30

**Decision:**

Major Revisions

**Comment:**

Please address the reviewer comments and resubmit